

# Particle export fluxes to the oxygen minimum zone of the Eastern Tropical North Atlantic

Anja Engel[1], Hannes Wagner[1], Frédéric A. C. Le Moigne[1], Samuel T. Wilson[2]

[1] GEOMAR Helmholtz Centre for Ocean Research Kiel,
24105 Kiel, Germany

[2] Daniel K. Inouye Center for Microbial Oceanography: Research and Education, Department of Oceanography, University of Hawaii, Honolulu, HI 96822, USA

*Correspondence to*: Anja Engel (aengel@geomar.de)

**Abstract.** In the ocean, sinking of particulate organic matter (POM) drives carbon export
from the euphotic zone and supplies nutrition to mesopelagic communities, the feeding and
degradation activities of which in turn lead to export flux attenuation. Oxygen ($O_2$) minimum
zones (OMZs) with suboxic water layers (<5 μmol $O_2$ $kg^{-1}$) show a lower carbon flux
attenuation compared to well oxygenated waters (>100 μmol $O_2$ $kg^{-1}$), supposedly due to
reduced heterotrophic activity. This study focuses on sinking particle fluxes through hypoxic
mesopelagic waters (<60% μmol $O_2$ $kg^{-1}$); these represent ~100 times more ocean volume
globally compared to suboxic waters, but have less been studied. Particle export fluxes and
attenuation coefficients were determined in the Eastern Tropical North Atlantic (ETNA) using
two surface tethered drifting sediment trap arrays with 7 trapping depths located between 100
and 600m. Data on particulate matter fluxes were fitted to the normalized power function
$F_z=F_{100} (z/100)^{-b}$, with $F_{100}$ being the flux at a depth (z) of 100 m and *b* being the attenuation
coefficient. Higher *b*-values suggest stronger flux attenuation and are influenced by factors
such as faster degradation at higher temperatures. In this study, *b*-values of organic carbon
fluxes varied between 0.74 and 0.80 and were in the intermediate range of previous reports,
but lower than expected from seawater temperatures within the upper 500m.  During this
study, highest *b*-values were determined for fluxes of particulate hydrolysable amino acids
(PHAA), followed by particulate organic phosphorus (POP), nitrogen (PN), carbon (POC),
chlorophyll *a,* and transparent exopolymer particles (TEP), pointing to a sequential
degradation of organic matter components during sinking. Our study suggests that in addition
to $O_2$ concentration, organic matter composition co-determines transfer efficiency through the
mesopelagial. The magnitude of future carbon export fluxes may therefore also depend on
how organic matter quality in the surface ocean changes under influence of warming,
acidification, and enhanced stratification.


## 1. Introduction

The biological carbon pump, defined as the export of biologically fixed carbon dioxide ($CO_2$) from the surface to the deeper ocean mainly in the form of sinking particles (Volk and Hoffert, 1985), influences atmospheric $CO_2$ concentration and affects ecosystem structure and elemental distributions in the ocean. The total amount of carbon export as well as the efficiency of the biological carbon pump, *i.e.* the ratio between export and primary production, are highly dynamic (Buesseler and Boyd, 2009; Lam et al., 2011). Changes in the efficiency of the biological carbon pump may have been responsible for past atmospheric $CO_2$ variability between glacial-interglacial transition periods (Kohfeld and Ridgewell, 2009) and play a key role for future climate predictions (Heinze et al., 2015).

Most of the POM being exported below the surface mixed layer (<200m in general) is solubilized and remineralized within the mesopelagic layer, *i.e.* between depths of 200 and 1000 m (Bishop et al., 1978; Suess, 1980). The shallower the carbon remineralization depth, the more likely is $CO_2$ to exchange with the atmosphere, and hence drive a shorter carbon storage time in the ocean (Volk and Hoffert, 1985; Kwon et al., 2009). Factors driving export flux attenuation in the mesopelagic have therefore a large influence on $CO_2$ sequestration in the ocean. The vertical profile of sinking particulate organic carbon (POC) flux has often been described by a normalized power function: $F_z = F_{100}(z/100)^{-b}$, where $F_z$ is the particle flux as a function of depth $z$, $F_{100}$ is the flux at 100 m depth, and $b$ is the flux attenuation coefficient (Martin et al., 1987; hereafter *M87*). The authors of the *M87* study derived an 'open ocean composite' for POC export fluxes from North Pacific data with a $F_{100} = 50.3$ mg m$^{-2}$ d$^{-1}$ and $b$ = 0.86. However strong regional variations of both total export POC fluxes and $b$ values are observed (Martin et al., 1987; Buesseler et al., 2007a; Torres Valdes et al., 2014; Marsay et al., 2015) with several factors proposed to control export flux attenuation. Increased attenuation, *i.e.* higher $b$-values, have been related to increased temperature (Marsay et al.,

2015), zooplankton feeding activity (Lampitt et al., 1990), coprophagy, coprorhexy, and coprochaly (Belcher et al. 2016), microbial cycling (Giering et al., 2014) and lack of ballast (LeMoigne et al., 2012). Decreased flux attenuation, *i.e*. lower *b*-values, and thus higher transfer efficiencies ($T_{eff}$) have been associated to high particle sinking velocity depending on plankton community composition, especially the presence of larger phytoplankton cells (Buesseler, 1998; Buesseler and Boyd 2009), particle aggregates (Alldredge and Gotschalk, 1989), and fecal pellets (Cavan et al., 2015). Organic polymers, such as transparent exopolymer particles (TEP) increase the rate of aggregate formation due to their high stickiness (Alldredge et al., 1993; Engel, 2000; Passow, 2002; Chow et al., 2015) and supposedly play an important role in particle export fluxes (Passow, 2002; Arrigo, 2007; Chow et al., 2015). TEP are carbon-rich particles that form from dissolved polysaccharides (Engel et al., 2004). When included in sinking POM inventories, TEP may increase carbon relative to nitrogen export fluxes, a mechanism potentially counteracting rising $CO_2$ concentration in the atmosphere (Schneider et al., 2004; Arrigo, 2007; Engel et al., 2014). However, TEP themselves are non-sinking due to a high water content and low density (Azetzu-Scott and Passow, 2004), and little quantitative data are available on TEP export by sinking particles so far (Passow et al., 2000; Martin et al., 2011; Ebersbach et al., 2014). Thus, the role of TEP in carbon export is still unresolved.

Reduced POC flux attenuation has also been suggested for oxygen minimum zones (OMZs) (Martin et al., 1987; Haake et al., 1992; Devol and Hartnett, 2001; Van Mooy et al., 2002; Keil et al., 2015) as a consequence of reduced zooplankton feeding and microbial degradation activities in suboxic (<5 μmol $O_2$ $kg^{-1}$) waters. So far, the vast majority of mesopelagic downward POM flux measurements originate from well oxygenated waters (>100 μmol $O_2$ $kg^{-1}$). In the *M87* study, five sets of drifting sediment traps were deployed in the oxygenated North Pacific and four sets were deployed in the Eastern Tropical North Pacific (ETNP) OMZ. The flux attenuation coefficients (*b*) for the oxygenated North Pacific averaged 0.90 ±

0.06, while lower $b$ values averaging 0.66 ± 0.24 were measured in the ETNP OMZ. In agreement, Devol and Hartnett (2001) and Van Mooy et al. (2002) observed low particle attenuation in the OMZ of the ETNP off Mexico, yielding $b$ coefficients of 0.36 and 0.40, respectively. Keil et al. (2015) found $b$ values of 0.59-0.63 in the suboxic Arabian Sea. These studies thus indicate that a greater proportion of the sinking POM escapes degradation while sinking through suboxic waters. However, influence of oxygen on organic matter degradation may vary between individual components. For instance, degradation of hydrolysable amino acid under suboxic conditions was found to continue with the same rate as compared to oxic conditions (Van Mooy et al. 2002; Pantoja et al. 2004), suggesting that anaerobic and micro-aerobic bacteria preferentially utilize nitrogen-rich components.

So far, little is known on sinking POM flux attenuation in hypoxic waters (<60 μmol $O_2$ $kg^{-1}$), which are more widespread (~4% of ocean volume) compared to suboxic waters (< 0.05% of ocean volume). Laboratory studies indicated that particle aggregates sinking through hypoxic waters can become suboxic within their interior due to oxygen diffusion limitation and evolve microbial degradation processes typical for suboxic waters (Alldredge and Cohen, 1987; Ploug et al., 1997; Stief et al., 2016). For example, at an ambient $O_2$ concentration of 60 μmol $kg^{-1}$, the $O_2$ uptake by a 2 mm (diameter) aggregate was diffusion-limited and a 0.5 mm wide anoxic core occurred within its interior (Ploug and Bergkvist, 2015). Since OMZs are expected to expand in the future as a consequence of global warming and altered circulation patterns (Stramma et al., 2008), the role of oxygen in controlling the biological pump efficiency needs to be better constrained for predicting ocean-climate feedbacks. In order to assess what controls carbon flux attenuation and depth-related changes in sinking particle composition in hypoxic waters, we determined downward POM fluxes in the ETNA off the coast of Mauritania, which exhibits an extensive hypoxic OMZ between 300 and 500 m. We used two parallel drifting, surface-tethered sediment trap devices with particle interceptor

traps (PITs) at 7-8 different depths between 60-600 m to estimate fluxes to and within the
OMZ.


**2. Methods**
2.1. The Study area
The study was conducted from March 17[th] to April 16[th] 2014 during a cruise of the RV
METEOR to the ETNA region off the coast of Mauritania (Fig. 1a). The study area included
hypoxic waters with minimum values of oxygen concentration of 40 µmol kg$^{-1}$ as determined
by CTD (Seabird) casts with two calibrated oxygen sensors at midwater depths of 350-500 m
(Fig. 1b) (Visbeck, 2014).

2.2. Sediment trap operation and sample analysis
Free-drifting surface-tethered sediment trap devices were deployed for 196 h during the first
deployment and 281h during the second deployment (Fig. 1c). The first trap device was
deployed on the 24[th] of March 2014 (11:00 UTC) at 10.00°N 21.00°W with 12 Particle-
Interceptor-Traps (PITs) at each of 8 depths: 60, 100, 150, 200, 300, 400, 500, and 600 m.
The device was recovered on the 1[st] of April 2014 (14:30 UTC) at 10.46°N 21.39°W. The
second device was deployed on the 27[th] of March 2014 (16:00 UTC) at 10.25°N 21°W with
12 PITs at each of 7 depths: 100, 150, 200, 300, 400, 500, and 600 m. The second trap device
was recovered on the 8[th] of April 2014 (09:00 UTC) at 10.63°N 21.50°W. Both devices
slowly drifted northwest and were recovered approximately 37 nm away from their
deployment location (Fig. 1c). Within the drifting area oxygen concentration in the OMZ
resembled the overall pattern of the Mauritanian upwelling with fully hypoxic conditions
between 300 and 500 m (Fig. 1d).
The design of the trap devices and the drifting array basically follows Knauer et al. (1979),
with 12 PITs mounted on a polyvinylchloride (PVC) cross frame. The PITs were acrylic tubes
with an inside diameter of 7 cm, an outside diameter of 7.6 cm and a height of 53 cm, leading
to an aspect ratio of 7.5. The aspect ratio and a baffle system consisting of smaller acrylic
tubes attached to the top end of each PIT help to reduce drag-induced movement within the
trap (Soutar et al., 1977). PVC crosses with PITs were attached to a free-floating line, which
was buoyed at the surface and weighed at the bottom. The surface buoys of the arrays carried
GPS/Iridium devices and flashlights.
Prior to each deployment, each PIT was filled with 1.5 L filtered surface seawater (0.2 μm
pore size cartridge) collected from the ship's underway seawater system, up to 3/4 of the
PITs' height. A brine solution was prepared by dissolving 50 g $L^{-1}$ sodium chloride with
filtered surface seawater and subsequently filtered through a 0.2 μm cartridge to remove
excess particulates. 20 ml of formalin was then added per L of the solution to achieve a brine
solution with 2% formalin. The preservative solution was then slowly transferred into each
PIT beneath the 1.5 L of filtered seawater using a peristaltic pump. PITs were covered with
lids immediately, to minimize contamination before deployment.
Sample treatment after trap recovery followed recommendations given by Buesseler et al.
(2007b). After recovery, all PITs were capped to minimize contamination. The density
gradient was visually inspected and found intact at the position of prior to deployment or at a
maximum 2 cm above. Then, seawater was pumped out of each PIT using a peristaltic pump
down to 2-3 cm above the density gradient. The remaining ~0.6 L were subsequently
transferred to canisters, pooled from 11 tubes per depth. 40 ml formalin were added to each
canister. Samples from each depth were passed through a 500 μm nylon mesh. Swimmers
were removed from the mesh with forceps under a binocular microscope and the remaining
particles, which stuck to the mesh, were transferred back to the sample. Samples were
subsequently split into aliquots of the total sample. Therefore, the pooled sample was
transferred into a round 10 L canister and stirred at medium velocity with a magnetic bar.
Aliquots were transferred into 0.5 L Nalgene bottles with a flexible tube using a peristaltic
pump. Aliquots of samples were filtered under low pressure (<200 mbar) onto different filter
types (combusted GF/F 0.7 μm, polycarbonate 0.4 μm, or cellulose acetate 0.8 μm; see below)
for different analyses and stored frozen (-20 °C) until analyses.

2.2.1. Biogeochemical Analyses
The following parameters were determined: Total particulate mass (TPM), particulate organic
carbon (POC), particulate nitrogen (PN), particulate organic phosphorus (POP), biogenic
silica (BSi), chlorophyll *a* (Chl *a*), particulate hydrolysable amino acids (PHAA) and
transparent exopolymer particles (TEP).

TPM was analyzed in triplicate. The following aliquots were filtered in triplicate onto pre-
weighed 0.4 μm polycarbonate filters: 800 ml (2 x 400 ml; 8 % of total sample) for the depths
of 600 m to 300 m of deployment #1, 400 ml (4 % of total sample) for the depths of 200 m
and 150 m of deployment #1 and for all depths of deployment #2, 420 ml (4 % of total
sample) for the depth of 100 m and 60 m of deployment #1. Filters were rinsed two times
with Milli-Q water, dried at $60^{\circ}$C for 4 h and stored until weight measurement on a Mettler
Toledo XP2U microbalance.

POC and PN aliquots were filtered in triplicate onto combusted (8h at 500°C) GF/F filters
(Whatmann, 25 mm): 400 ml (4 % of total sample) for the depths of 600 m to 150 m of
deployment #1, 420 ml (4 % of total sample) for the depths of 100 m and 60 m of deployment
#1, 100 ml (1 % of total sample) for all depths of deployment #2. For the depths of 150 m,
100 m and 60 m of deployment #1, 400 - 420 ml (4 % of total sample) was filtered onto two
filters, due to the high particle load at these depths. Filters were exposed to fuming
hydrochloric acid in a fuming box over night to remove carbonate and subsequently dried
(60°C, 12 h). For analysis, the filters were enclosed in tin cups and analysed using an Euro
EA elemental analyzer calibrated with an acetanilide standard. For the depths of 150, 100 and
60 m of deployment #1 the sum of both filters was taken.

POP was determined in triplicate, except for 60 m depth of deployment #1, which was only
determined in duplicate. The following aliquots were filtered onto combusted GF/F filters
(Whatmann, 25 mm): 400 ml (4 % of total sample) for the depths of 600 m to 150 m of
deployment #1, 420 ml (4 % of total sample) for the depths of 100 m and 60 m of deployment
#1, 100 ml (1 % of total sample) for all depths of deployment #2. For the depths of 200 m to
60 m of deployment #1, the volume of 400 ml/ 420 ml (4 % of total sample) was filtered onto
two filters, due to the high particle load at these shallower depths. Organic phosphorus
collected on the filters was digested in the potassium peroxydisulphate-containing substance
Oxisolv (Merck) for 30 min in a pressure cooker and measured colorimetrically as ortho-
phosphate following the method of Hansen and Koroleff (1999).

PHAA were determined in duplicate. The following aliquots were filtered onto combusted
GF/F filters (25 mm): 400 ml (4 % of total sample) for the depths of 600 m to 150 m of
deployment #1, 420 ml (4 % of total sample) for the depths of 100 m and 60 m of deployment
#1, 100 ml (1 % of total sample) for all depths of deployment #2. For the depths of 150 m,
100 m and 60 m of deployment #1, the volume of 400 ml / 420 ml (4 % of total sample) was
filtered onto two filters, due to the high particle load at these shallower depths. PHAA
analysis was performed according to Lindroth & Mopper (1979) and Dittmar et al. (2009)
with some modifications. Duplicate samples were hydrolyzed for 20 h at 100°C with
hydrochloric acid (30%, Suprapur, Merck) and neutralized by acid evaporation under vacuum
in a microwave at 60°C. Samples were washed with water to remove remaining acid.
Analysis was performed on a 1260 HPLC system (Agilent). Thirteen different amino acids
were separated with a C18 column (Phenomenex Kinetex, 2.6 µm, 150 x 4.6 mm) after in-line
derivatization with o-phtaldialdehyde and mercaptoethanol. The following standard amino
acids were used: aspartic acid (AsX), glutamic acid (GlX), histidine (His), serine (Ser),
arginine (Arg), glycine (Gly), threonine (Thr), alanine (Ala), tyrosine (Tyr), valine (Val),
phenylalanine (Phe), isoleucine (Ileu), leucine (Leu), γ- amino butyric acid (GABA). α-
amino butyric acid was used as an internal standard to account for losses during handling.
Solvent A was 5% acetonitrile (LiChrosolv, Merck, HPLC gradient grade) in
sodiumdihydrogenphospate (Merck, suprapur) buffer (pH 7.0), Solvent B was acetonitrile. A
gradient was run from 100% solvent A to 78% solvent A in 50 minutes. The detection limit
for individual amino acids was 2 nmol monomer $L^{-1}$. The precision was <5%, estimated as the
standard deviation of replicate measurements divided by the mean. The degradation index
(DI) was calculated from the amino acid composition following Dauwe et al. (1999).

BSi was determined in triplicate. The following aliquots were filtered onto cellulose acetate
filters (0.8 µm): 400 ml (4 % of total sample) for the depths of 600 m to 150 m of deployment
#1, 420 ml (4 % of total sample) for the depths of 100 m and 60 m of deployment #1, 200 ml
(2 x 100 ml; 2 % of total sample) for all depths of deployment #2. Filters were incubated with
25 ml NaOH (0.1 M) at 85˚C for 2h 15min in a shaking water bath. After cooling of the
samples, analysis was conducted according to the method for determination of $Si(OH)_4$ by
Hansen and Koroleff (1999). Fluxes of biogenic opal were calculated assuming a water
content of ~10% and therefore the chemical formula $SiO_2$ x $0.4H_2O$ with a density of ~2.1 g
$cm^{-3}$ (Mortlock and Fröhlich 1989).

Chl *a* was determined in duplicate. The following aliquots were filtered onto GF/F filters (25
mm): 400 ml (4 % of total sample) for the depths of 600 m to 150 m of deployment #1, 420
ml (4 % of total sample) for the depths of 100 m and 60 m of deployment #1, 100 ml (1 % of
total sample) for all depths of deployment #2. For the depths of 200 m to 60 m of deployment
#1, the volume of 400 ml / 420 ml (4 % of total sample) was filtered onto two filters, due to
the high particle load at these shallower depths. Samples were analyzed after extraction with
10ml of acetone (90%) on a Turner fluorimeter after Welschmeyer (1994). Calibration of the
instrument was conducted with spinach extract standard (Sigma Aldrich).

TEP were determined in quadruplet by microscopy after Engel (2009). Between 3.5 and 10 ml
(0.03-0.1% of total sample) for the depths of deployment #1 and #2 were filtered onto 0.4 µm
Nuclepore membrane filters (Whatmann) and  stained with 1 mL Alcian Blue solution. Filters
were mounted onto Cytoclear© slides and stored at -20 °C until microscopy analysis using a
light microscope (Zeiss Axio Scope A.1) connected to a camera (AxioCAM Mrc). Filters
were screened at 200x magnification. 30 pictures were taken randomly from each filter in two
perpendicular cross sections (15 pictures each; resolution 1040 x 1040 pixel, 8-bit color
depth). Image analysis software WCIF ImageJ (Version 1.44, Public Domain, developed at
the US National Institutes of Health, courtesy of Wayne Rasband, National Institute of Mental
Health, Bethesda, Maryland) was used to semi-automatically analyse particle numbers and
area.

The carbon content of TEP (TEP-C) was estimated after Mari (1999) using the size dependent
relationship:

$\text{TEP-C} = a \sum_i (n_i\ r_i^D)$,                                                (1)

with $n_i$ being the number of TEP in the size class i and $r_i$ the mean equivalent spherical radius
of the size class. The constant $a = 0.25 * 10^{-6}$ (µg C) and the fractal dimension of aggregates
D= 2.55 were proposed by Mari (1999). TEP-C was only calculated for the size fraction <5
μm including mainly free TEP, because larger TEP included TEP covered aggregates with
solid particles. Estimating carbon content of these larger particles would overestimate TEP-C
as the volume of the other particles would be included.

2.3. Calculations and statistics
Fluxes of $CaCO_3$ and lithogenic matter (lith) were calculated as:

$[CaCO_3 + lith] = [TPM]-[POM]-[Opal]$,                    (2)

Total mineral ballast (ballast$_{total}$) was calculated as:

$[ballast_{total}] = [TPM]-[POM]$,                    (3)

and the percentage of ballast$_{total}$ (%ballast$_{total}$ ) was calculated as:

$[\%ballast_{total}] = ([TPM] - [POM])/[TPM]*100$,                    (4)

The transfer efficiency ($T_{eff}$) of particulate components was calculated as the ratio of fluxes at
600 m to those at 100 m.

Calculated mean values include replicate measurements of both deployments. Data fits and
statistical tests were performed with the software packages Microsoft Office Excel 2010,
Sigma Plot 12.0 (Systat) and Ocean Data View (ODV) (Schlitzer, 2013). Weighted-average
gridding was used in ODV to display data according to data coverage with automatic scale
lengths. The overall significance level was $p<0.05$.


## 3. Results and Discussion


3.1. Fluxes of different compounds

Export fluxes of TPM and particulate organic elements determined during both trap deployments showed good overall agreement and a decrease with depth, fitting well to the power law function of *M87* (Fig. 2a-d, Fig.3a-d and Table 1). Averaging fluxes from both deployments yielded a total mass flux of 240 $\pm$ 34 mg m$^{-2}$ d$^{-1}$ at 100 m decreasing to 141 $\pm$ 8.8 mg m$^{-2}$ d$^{-1}$ in the core of the OMZ (400 m) (Fig. 2a). Fluxes of POC, PN and POP at 100 m depth were 73 $\pm$ 8.8, 13$\pm$ 1.4 and 0.67 $\pm$ 0.06 mg m$^{-2}$ d$^{-1}$, respectively, and decreased to 26 $\pm$ 4.5, 3.0 $\pm$ 0.41 and 0.19 $\pm$ 0.04 mg m$^{-2}$ d$^{-1}$ at 400 m depth (Fig. 2b-d). The contribution of POC flux to total mass flux (% OC) decreased from about 30% at 60-150 m depth to 17-20% at 400 m depth and showed only a minor decrease below 400 m, to 14-16% at 600 m depth. Similarly, the percentage of PN flux to total mass flux (% N) showed the largest decrease between 60 and 400 m, i.e. from 6.6% to 2.0-2.3%, and less decline below, reaching 1.7-1.8% at 600 m. The percentage of POP flux to total mass flux (% P) decreased from 0.37% at 60 m depth to 0.11-0.16% at 400 m depth, and remained constant below 400 m depth. No previous data are available for POM export fluxes at our study site for direct comparison. However, our trap data compare well to carbon export fluxes estimated from particle size data (i.e. 10-300 mg C m$^{-2}$ d$^{-1}$) reported for 100 m depth in the area off Cape Blanc (Mauritania) by Iversen et al. (2010).


Fluxes of phytoplankton biomass, as indicated from Chl *a*, were similar at 100 m during both deployments, with 104 $\pm$ 1.5 µg Chl *a* m$^{-2}$ d$^{-1}$ during the first and 116 $\pm$ 6.2 µg m$^{-2}$ d$^{-1}$ during the second deployment, but behaved differently below, with a stronger flux attenuation above

the OMZ during the first compared to the second deployment (Fig. 3a). Fluxes within the
OMZ core were $35 \pm 0.1$ µg m$^{-2}$ d$^{-1}$ (#1) and $53 \pm 0.5$ µg m$^{-2}$ d$^{-1}$ (#2) respectively.

Opal fluxes were also similar during both deployments, yielding an average of $47 \pm 3.6$ mg m$^{-2}$
d$^{-1}$ at 100 m, steadily decreasing to $32 \pm 2.4$ mg m$^{-2}$ d$^{-1}$ at 400 m depth (Fig. 3b). Similar to
Chl *a*, opal fluxes were slightly higher above the OMZ during the second compared to the
first deployment, but quite similar or even lower below the OMZ. This may indicate that the
second trap device, which drifted more northerly (Fig. 1c), exploited waters of a more recent
diatom bloom compared to the first deployment.
Fluxes of [CaCO$_3$ + lith] were similar to opal fluxes during the first deployment (F$_{100}$=52 mg
m$^{-2}$ d$^{-1}$) but considerably lower during the second (F$_{100}$=14.8 mg m$^{-2}$ d$^{-1}$) (data not shown).

During this study, export fluxes of TEP were estimated from decrease over depth of total
particle area and showed the strongest depth attenuation between 60 and 100 m during the
first deployment (Fig. 3c). Like Chl *a* fluxes, TEP export fluxes were slightly higher during
the second compared to the first deployment. At 100 m depth, average TEP flux was $1860 \pm$
46 cm$^2$ m$^{-2}$ d$^{-1}$ and decreased to $1190 \pm 52$ cm$^2$ m$^{-2}$ d$^{-1}$ at 400 m. Using a TEP size to carbon
conversion according to Mari (1999) yielded to an average TEP-C (<5 µm) flux of $1.73 \pm$
0.35 mg C m$^{-2}$ d$^{-1}$ at 100m depth, slightly decreasing to $1.64 \pm 0.28$ mg m$^{-2}$ d$^{-1}$ at 400 m and
further to $0.90 \pm 0.32$ mg m$^{-2}$ d$^{-1}$ at 600 m. Although TEP supposedly play an important role
in particle export fluxes (Passow, 2002; Arrigo, 2007; Chow et al., 2015), only a few previous
estimates for TEP export fluxes based on sediment traps have been given so far to which we
can compare our data. Martin et al. (2011) measured TEP export fluxes during a spring
bloom in the Iceland Basin (Northeast Atlantic Ocean) using the PELAGRA neutrally
buoyant sediment traps and determined values in the range of 30-120 mg Gum Xanthan
Equivalent m$^{-2}$ d$^{-1}$. Ebersbach et al. (2014) obtained lower values of 0.03-5.14 mg Gum
Xanthan Equivalent $m^{-2}$ $d^{-1}$ during the LOHAFEX iron fertilization experiment in the
Southern Ocean. Assuming a conversion factor of 0.63 µg C µg$^{-1}$ Gum Xanthan after Engel
(2004), these previous estimates suggest TEP-C export fluxes ranging from 0.02 to 3 mg $m^{-2}$
$d^{-1}$ for the Southern Ocean and from 19 to75 mg $m^{-2}$ $d^{-1}$ for the North Atlantic spring bloom.
Our data on TEP export fluxes for ETNA region are within the range of these previous
studies, but closer to the lower estimates for the Southern Ocean. It has to be emphasized,
though, that our calculated TEP-C fluxes are likely underestimates, since only suspended, i.e.
'free' TEP < 5 µm were taken into account. TEP-C associated to aggregates cannot be
determined with the applied microscopic technique. Overall, TEP-C export fluxes in the
ETNA were significantly related to Chl *a* fluxes, yielding [TEP-C, mg $m^{-2}$ $d^{-1}$] = 11.9 [Chl *a*;
mg $m^{-2}$ $d^{-1}$] + 0.74 ($r^2$=0.59, n= 15, p<0.01).

A strong decrease at shallow depth (60–100 m) was also observed for PHAA fluxes during
the first deployment (Fig. 3d). Average PHAA fluxes were 330 ± 51 µmol $m^{-2}$ $d^{-1}$ at 100 m,
and 90 ± 20 µmol $m^{-2}$ $d^{-1}$ in the OMZ core at 400 m. These fluxes are equivalent to amino
acid related fluxes of 16.8 ± 2.6 mg C $m^{-2}$ $d^{-1}$ (100 m) and 4.48± 1.0 mg C $m^{-2}$ $d^{-1}$ (400 m),
respectively, which are typical values for PHAA-C fluxes in the ocean (Lee and Cronin,
1984). PHAA fluxes decreased slightly within the OMZ, i.e. from 300 to 500 m.


3.2.   Flux attenuation in the ETNA OMZ
Fluxes from both deployments were fitted to the exponential decrease model (Martin et al.,
1987) and attenuation coefficients (*b*-values) were estimated for all components (table 1).
Higher *b*-values suggest stronger attenuation and may hint to either reduced sinking velocities
of particles or to faster degradation of more labile components. During this study, PHAA
were the most rapidly attenuated components of sinking particles, followed by POP, PN,
POC, Chl *a,* and TEP (table 1). Attenuation of mineral fluxes was less pronounced than for
TPM.
Attenuation coefficient of POC export fluxes was 0.80 during the first and 0.74 during the
second deployment. These values are in the intermediate range of previously determined *b*-
values for POC attenuation in the mesopelagic, shown to vary between 0.51 as determined in
the North Pacific (K2) and 1.59 as determined for the NASG (Buesseler et al., 2007a; Marsay
et al., 2015). Based on trap data from fully oxygenated water columns, Marsay et al. (2015)
recently suggested a linear relationship between POC flux attenuation and median water
temperature within the upper 500m of the water column according to: $b=0.062T+0.303$.
Applying this relationship to our study area, with temperature decreasing from 26°C at the
surface to 9°C at 500 m and a median temperature value of 12.01°C, would give a *b*-value of
1.05. This estimated *b*-value is higher than the values observed in this study (0.74 - 0.80) and
suggests that oxygen deficiency may reduce attenuation of POC fluxes in the ETNA resulting
in higher $T_{eff}$ of organic matter though the OMZ's compared to well oxygenated waters.

Differences in flux attenuation coefficients translate into different $T_{eff}$ for individual
components, with PHAA being the least and TEP being the most efficiently exported organic
component (table 1). In particular, values of $T_{eff}$ for TEP and therewith for TEP-C were about
three times higher than for PHAA-C and even clearly higher than for bulk POC, suggesting a
preferential export of carbon included in TEP below 100 m.   However, a steep decrease of
TEP flux was observed between 60 m and 100 m during the first deployment. TEP are
produced by a variety of organisms, i.e. different phytoplankton and bacterial species and
cannot be considered as of homogenous composition. Several mechanisms may therefore be
responsible for a change in TEP transfer efficiency with depth: 1) change of TEP
degradability with depth, 2) differences in TEP composition over depth related to association
with particles of different settling speed, 3) new production of TEP, abiotically or by bacteria,
during solubilization and degradation of sinking particles, 4) capture of suspended TEP by
sinking aggregates, or 5) reduced degradation rate of TEP at lower oxygen. In support of the
latter hypothesis, an attenuation of TEP fluxes within the OMZ (300-500 m) was not
detectable, but rather occurred again below the OMZ.

3.3. Changes in POM composition during export
POM, assumed to be $2.2 \times$ [POC] following Klaas and Archer (2002) made the greatest
contribution to TPM flux at 60 m, but decreased below. Conversely, [%ballast$_{total}$] increased
with depth, namely from 30% w/w at 60 m to 68% w/w at 600 m.
Biogenic opal (density: 2.1 g cm$^{-3}$) in the ocean is produced mainly by diatoms and
radiolarians. During this study, opal made a rather constant contribution to TPM fluxes with
20-25% weight below 100 m. Hence, the observed increase in the [%ballast$_{total}$] with depth
was due to an increasing contribution of $CaCO_3$ and lithogenic material. [$CaCO_3$+ lith] to
TPM increased from 10-15% above 150 m to 45% at 600 m. As a consequence, the ballast
ratio, defined as [Opal]:[$CaCO_3$+lith] changed from a dominance of opal above the OMZ to a
dominance [$CaCO_3$+lith] within and below the OMZ (Figure 4). Slight differences were
observed between the two deployments. Contribution of opal and of [$CaCO_3$+lith] to TPM at
100m was almost equal during the first deployment with a share of 18% and 22%,
respectively. During the second deployment the contribution of opal to TPM at 100 m was
21% but only 6% for [$CaCO_3$+lith]. Thus, the higher contribution of opal to TPM fluxes
together with higher Chl *a* fluxes indicated that diatomaceous material had a higher share of
particles sinking out of the euphotic zone down to the OMZ core during the second compared
to the first deployment.

Molar [POC]:[PN] ratios were close to the Redfield ratio at depths shallower than 100 m,
increased to a ratio of 10 at 400 m depth and remained  constant between 400 and 600 m
depth (Fig. 5a). [PN]:[POP] ratios were much above Redfield, with values varying between
30 and 45 throughout the water column (Fig. 5b). Also [POC]:[POP] ratios were much higher
than Redfield ratios, and showed an increasing trend down to 300-400 m depth, while
decreasing below (Fig. 5c). These changes in elemental ratios suggested a preferential
remineralization of POP in the upper 300 m, followed by PN and POC deeper down.
The percentage of total organic matter in TPM fluxes decreased from 67% at 100m to 32% at
600m (Fig. 6d). As a consequence of higher $T_{eff}$ of TEP relative to bulk POC, contribution of
TEP-C to POC increased significantly with depth during both deployments (p<0.01; r²=0.59,
n=15) and was 2% at 100 m, and 6% within and 5% below the OMZ (Fig. 5e). Because TEP
do not sink by themselves their export to depth depends on their incorporation into settling
aggregates. In a laboratory study, Engel et al. (2009) observed that decomposition of TEP was
faster relative to bulk POC for aggregates formed from calcifying and non-calcifying
*Emiliania huxleyi* cultures. In that experiment, aggregate decomposition was investigated
under oxic conditions. Other studies also showed fast microbial degradation of TEP under
oxic conditions (Bar-Zeev and Rahav, 2015). One possible explanation for increasing [TEP-
C]:[POC] in the hypoxic OMZ of the ETNA region could be that TEP are mostly included in
sinking aggregates, whereas POC could be included in various particle types, such as large
cells, detritus or fecal pellets. Ploug et al. (1997) estimated that carbon turn-over time inside
anoxic aggregates can be strongly reduced. Due to high microbial activity and reduced water
exchange aggregates sinking into hypoxic waters are more likely to experience anoxic
conditions than individual particles (Ploug and Bergkvist, 2015). Thus, TEP settling into
hypoxic waters by aggregates may be exposed to anoxia, and therewith to reduced microbial
degradation, in consequence leading to a preferential TEP transfer through the OMZ. This
may also explain the decrease of [TEP-C]:[POC] ratios below the OMZ at 600 m water depth,
which was, however, only observed during the second deployment. Since PN was more
rapidly degraded than POC this also implied that the ratio of [PN]:[TEP-C] became lower
with depth.

In contrast to [TEP-C]:[POC], values of [PHAA-C]:[POC] in POM fluxes declined during
both deployments above the OMZ. However, in the core of the OMZ, at 400 m, [PHAA-
C]:[POC] was higher than at 300 and 500 m (Fig. 5f); the same pattern was also observed
for [PHAA-N]:[PN] (data not shown). A faster decline in PHAA in sinking particles mainly
above but not within the OMZ is different to observations gained for more extensively
oxygen-deficient to full anoxic waters of the Eastern Tropical south Pacific (ETSP), which
suggested that PHAA are preferentially degraded under low oxygen conditions (Van Mooy et
al., 2002). In those studies, total hydrolysable amino acid (THAA) degradation under anoxic
conditions was found to continue with the same rate compared to oxic conditions, while
degradation of non-amino acid compounds was found to slow down (Pantoja et al., 2004; Van
Mooy et al., 2002). A preferential degradation of nitrogen-rich compounds over POC suggests
that microbes degrading organic matter under strongly oxygen deficient conditions via
denitrification preferentially utilize nitrogen-rich amino acids (Van Mooy et al., 2002). Our
data on PHAA do not suggest preferential amino acid loss due to components of sinking POM
degradation in the ETNA OMZ. This is in accordance with the absence of microbial N-loss
processes/ absence of denitrifying bacteria in ETNA oxygen deficient waters (Löscher et al.,
2016). Instead, a slight increase of [PHAA-C]:[POC] in the OMZ may point to higher protein
production by bacterial growth as previously observed for mesopelagic waters (Lee and
Cronin, 1982, 1984) and may be related to increased growth efficiency of bacteria
experiencing low oxygen condition as suggested by Keil et al. (2016).

Among all amino acids determined, GlX, Gly, Gaba and Leu showed the most pronounced
variations with depth (Fig. 6a-d, table 2). Whereas GlX and Leu showed a decrease with

depth (Fig. 6a, c), Gly continuously increased. It has been shown that Gly is enriched in the silica-protein complex of diatom frustules (Hecky et al., 1973). Preservation of frustules relative to POM may therefore explain the relative increase of Gly with depth in sinking particles. GlX has been used as a biomarker (Abramson et al., 2010), since GlX was shown to be enriched in calcerous plankton (Weiner and Erez, 1984). During this study %Mol of GlX was higher during the first deployment, which is in accordance with the observed higher contribution of [$CaCO_3$+lith] to TPM flux. Gaba has been used as an indicator for bacterial decomposition activity (Lee and Cronin, 1982; Dauwe and Middelburg, 1998; Engel et al., 2009). During this study %Mol Gaba behaved differently during the first compared to the second deployment with similar values within the OMZ, a pattern also observed for opal fluxes (Fig. 3b). Moreover, %Mol of Gaba showed a local peak at 300 m, i.e. within the upper oxycline, and may point to high bacterial activity at this depth. Leu is an essential amino acids and readily taken up by heterotrophic microorganisms. Little change in %Leu in the OMZ core (Fig. 3d) compared to above (<300 m) indicated reduced microbial reworking of organic matter under hypoxic conditions. Another indication of microbial reworking of organic matter can be derived from the Degradation index (DI) (Dauwe et al., 1999). During this study, the DI decreased with increasing depth, but with differences between the deployments (Fig. 7). During #2, DI was slightly higher above the OMZ indicating fresher material. During #1 DI did not decrease within the OMZ, but it continued to decrease from 300 m to 500 m depth during deployment #2. Together with observations on Chl *a* and opal fluxes, as well as changes in ballast ratio, data on DI suggest that the particles of more diatomaceous origin likely continued to decompose under hypoxic conditions.

**4. Conclusions**

Despite an improvement in understanding principle processes and drivers of particle export
processes over the past decades, spatial and temporal variability of export fluxes in the ocean
are still difficult to predict. This is partly due to the lack of observations in different regions
of the mesopelagic realm. Our study is the first to describe fluxes of POM in the hypoxic
mesopelagic waters of the ETNA. Our data suggest a higher transfer efficiency than expected
from seawater temperature solely, suggesting reduced degradation of organic matter by
heterotrophic communities at low oxygen concentration (<60 $\mu$mol $O_2$ $kg^{-1}$). The biological
carbon pump in high productivity regimes associated to OMZs, i.e. Eastern Boundary
Upwelling Systems such as the ETNA region off Mauritania, may therewith be more efficient
than in fully oxygenated waters of comparable temperature. In contrast to suboxic systems (<
5 $\mu$mol $O_2$ $kg^{-1}$) a relatively higher loss of amino acids from POM fluxes was not evident for
the hypoxic water-column, suggesting microbial N-loss processes were comparatively minor
within particles. This, however, requires further investigation since no corresponding rate
measurements of denitrification or anammox were conducted during this study. Organic
matter composition seems to have a large impact on transfer efficiencies as carbon fluxes
associated to amino acids were much more attenuated over depth than carbon fluxes
associated to polysaccharide-rich TEP. If these findings are transferable to other oceanic
regions, changes in surface ocean organic matter composition in response to climate change
may also impact the carbon remineralization depth and therewith may have a feed-back
potential to atmospheric $CO_2$ concentration that yet has to be assessed.

**5. Competing interest**
The authors declare that they have no conflict of interest.

**6. Acknowledgements**
This study is a contribution to the Collaborative Research Center 754 / SFB
Sonderforschungsbereich 754 'Climate-Biogeochemistry Interactions in the Tropical Ocean'.
We thank Martin Visbeck, Toste Tanhua, Tobias Hahn, Sunke Schmidtko, and Gerd
Krahmann for scientific and technical support as well as for providing oxygen and CTD data.
Many thanks go to the shipboard scientific party and crew of Meteor cruise M105. Jon Roa,
Ruth Flerus, Scarlett Sett and Tania Klüver are acknowledged for technical assistance. We
thank Cindy Lee (Stony Brook University) for helpful advices. FACLM is supported by the
DFG Excellence cluster Future Ocean. All data will become available at www.pangea.de
upon publication.

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

**Tables**

Table 1: Fluxes of particulate components at 100m depth ($F_{100}$) and in the core of the OMZ at
400m ($F_{OMZ}$), as well as the associated attenuation coefficients (*b*-values) and transfer

efficiencies ($T_{eff}$, %) over the depth range 100 to 600 m during two traps deployments in the ETNA. All units are in mg m$^{-2}$ d$^{-1}$ except for TEP fluxes which is reported in total particle area cm$^{-2}$ m$^{-2}$ d$^{-1}$. Mean values and standard deviations (SD) were calculated from analytical replicates.

| Component | | $F_{100}$ | | $F_{OMZ}$ | | b-value | | | $T_{eff}$ (%) |
|---|---|---|---|---|---|---|---|---|---|
| | | mean | SD | mean | SD | mean | SD | r² | (600/100 m) |
| Mass | I | 249 | 48.9 | 141 | 6.8 | -0.429 | 0.090 | 0.987 | 41 |
| | II | 231 | 16.3 | 141 | 12.1 | -0.355 | 0.033 | 0.998 | 52 |
| POC | I | 69.4 | 9.23 | 23.8 | 5.4 | -0.795 | 0.031 | 0.989 | 23 |
| | II | 76.3 | 8.43 | 28.1 | 3.0 | -0.741 | 0.044 | 0.989 | 22 |
| PN | I | 11.9 | 1.29 | 2.76 | 0.46 | -1.013 | 0.026 | 0.992 | 15 |
| | II | 13.5 | 1.12 | 3.26 | 0.19 | -1.00 | 0.020 | 0.990 | 16 |
| POP | I | 0.71 | 0.07 | 0.15 | 0.02 | -1.081 | 0.074 | 0.992 | 18 |
| | II | 0.64 | 0.03 | 0.22 | 0.02 | -0.80 | 0.034 | 0.990 | 23 |
| Opal | I | 44.6 | 1.76 | 34.0 | 1.7 | -.0195 | 0.038 | 0.987 | 65 |
| | II | 48.6 | 4.16 | 30.7 | 2.0 | -0.345 | 0.052 | 0.987 | 44 |
| Chl $a$ | I | 0.10 | 0.00 | 0.035 | 0.001 | -0.820 | 0.024 | 0.990 | 21 |
| | II | 0.12 | 0.01 | 0.053 | 0.005 | -0.625 | 0.082 | 0.988 | 24 |
| TEP | I | 1650 | 548 | 119 | 36.8 | -0.498 | 0.014 | 0.548 | 33 |
| | II | 2990 | 348 | 1644 | 95 | -0.451 | 0.069 | 0.810 | 37 |
| PHAA-C | I | 3.21 | - | 3.71 | 0.47 | -1.324 | 0.067 | 0.994 | 11 |
| | II | 1.28 | 0.10 | 5.24 | 0.79 | -0.978 | 0.096 | 0.991 | 14 |

Table 2: Composition (%Mol) and degradation index (DI) of PHAA collected at different depths during two trap deployments (#I, #II) in the ETNA
region.

| Depth(m) | AsX | GlX | Ser | Gly | Thr | Arg | Ala | GABA | Tyr | Val | Iso | Phe | Leu | DI |
|---|---|---|---|---|---|---|---|---|---|---|---|---|---|---|
| #I | | | | | | | | | | | | | | |
| 60 | 14.15 | 13.94 | 8.46 | 14.29 | 7.76 | 5.90 | 11.94 | 0.22 | 0.84 | 5.69 | 4.57 | 4.00 | 8.26 | 0.34 |
| 100 | 13.95 | 13.53 | 8.29 | 14.65 | 7.87 | 5.77 | 11.57 | 0.19 | 1.64 | 5.66 | 4.56 | 4.07 | 8.24 | 0.23 |
| 150 | 14.19 | 12.73 | 8.54 | 15.93 | 8.10 | 5.78 | 11.42 | 0.31 | 0.96 | 5.68 | 4.44 | 4.05 | 7.87 | 0.29 |
| 200 | 14.17 | 12.05 | 9.29 | 16.02 | 8.05 | 5.61 | 11.69 | 0.49 | 1.10 | 5.65 | 4.30 | 4.04 | 7.54 | 0.07 |
| 300 | 13.19 | 11.75 | 8.58 | 17.71 | 7.98 | 5.31 | 12.10 | 0.37 | 1.82 | 5.77 | 4.15 | 3.83 | 7.43 | 0.03 |
| 400 | 14.15 | 11.77 | 9.03 | 18.54 | 7.94 | 5.72 | 10.85 | 0.46 | 1.25 | 5.58 | 3.93 | 3.80 | 6.97 | 0.02 |
| 500 | 14.06 | 11.89 | 9.55 | 18.70 | 7.18 | 6.01 | 11.02 | 0.55 | 1.29 | 5.19 | 3.86 | 3.65 | 7.05 | 0.02 |
| 600 | 14.15 | 13.94 | 8.46 | 14.29 | 7.76 | 5.90 | 11.94 | 0.22 | 0.84 | 5.69 | 4.57 | 4.00 | 8.26 | 0.07 |
| #II | | | | | | | | | | | | | | |
| 100 | 13.89 | 14.69 | 8.36 | 12.94 | 7.57 | 5.89 | 12.26 | 0.21 | 0.02 | 6.13 | 5.12 | 4.05 | 8.86 | 0.24 |
| 150 | 13.48 | 14.23 | 8.46 | 14.12 | 7.56 | 5.68 | 12.55 | 0.22 | 0.00 | 6.21 | 5.01 | 3.85 | 8.62 | 0.37 |
| 200 | 13.80 | 13.90 | 9.10 | 14.27 | 7.20 | 6.12 | 11.57 | 0.27 | 0.04 | 6.19 | 5.07 | 3.97 | 8.49 | 0.13 |
| 300 | 14.58 | 14.63 | 8.35 | 15.16 | 7.75 | 5.56 | 11.75 | 0.26 | 0.14 | 5.62 | 4.51 | 3.82 | 7.88 | 0.07 |
| 400 | 14.06 | 13.01 | 8.72 | 16.45 | 7.99 | 5.55 | 11.74 | 0.44 | 0.79 | 5.54 | 4.33 | 3.77 | 7.59 | 0.08 |
| 500 | 14.08 | 12.90 | 8.75 | 16.48 | 7.59 | 5.69 | 11.81 | 0.37 | 0.30 | 5.94 | 4.62 | 3.80 | 7.66 | -0.09 |
| 600 | 13.62 | 12.55 | 9.16 | 17.02 | 7.95 | 5.75 | 11.23 | 0.42 | 0.38 | 5.87 | 4.61 | 3.88 | 7.55 | -0.04 |

**Figure captions:**


Figure 1a-d: Map of the study area (A) and depth distribution of oxygen concentration (mol
kg$^{-1}$) (B) in the Eastern Tropical North Atlantic (ETNA) during the RV Meteor 105 cruise,
when two surface tethered drifting sediment traps (STDT) were deployed (C). Depth
distribution of oxygen concentration (mol kg$^{-1}$) at stations visited in the deployment area
showed an oxygen minimum zone in the upper mesopelagial (D).

Figure 2a-d: Fluxes of total mass (a) and particulate organic carbon (b; POC), particulate
nitrogen (c; PN), and particulate organic phosphorus (d; POP) during the deployment of two
STDT in the ETNA. Deployments: Solid symbols #I, open symbols #II.

Figure 3a-d: Fluxes of Chlorophyll *a* (a; Chl *a*), opal (b), TEP (c), and PHAA (d) during the
deployment of two STDT in the ETNA. Deployments: Solid symbols #I, open symbols #II.

Figure 4: Changes in mineral ballast ratios of sinking particles with depth during the two
deployments in the ETNA. Deployments: Black bars #I, grey bars #II.

Figure 5a-f: Changes in organic matter composition of particles sinking through the OMZ
during the deployment of two STDT in the ETNA. Deployments: Solid symbols #I, open
symbols #II.

Figure 6a-d: Molar percentages of selected amino acids contained in PHAA during the
deployment of two STDT in the ETNA. Deployments: Solid symbols #I, open symbols #II.

Figure 7: Degradation index (DI) of organic matter in trap collected sinking particles based on
amino acid composition and calculated after Dauwe et al. (1999). Deployments: Black bars
#I, grey bars #II.



**Figures**

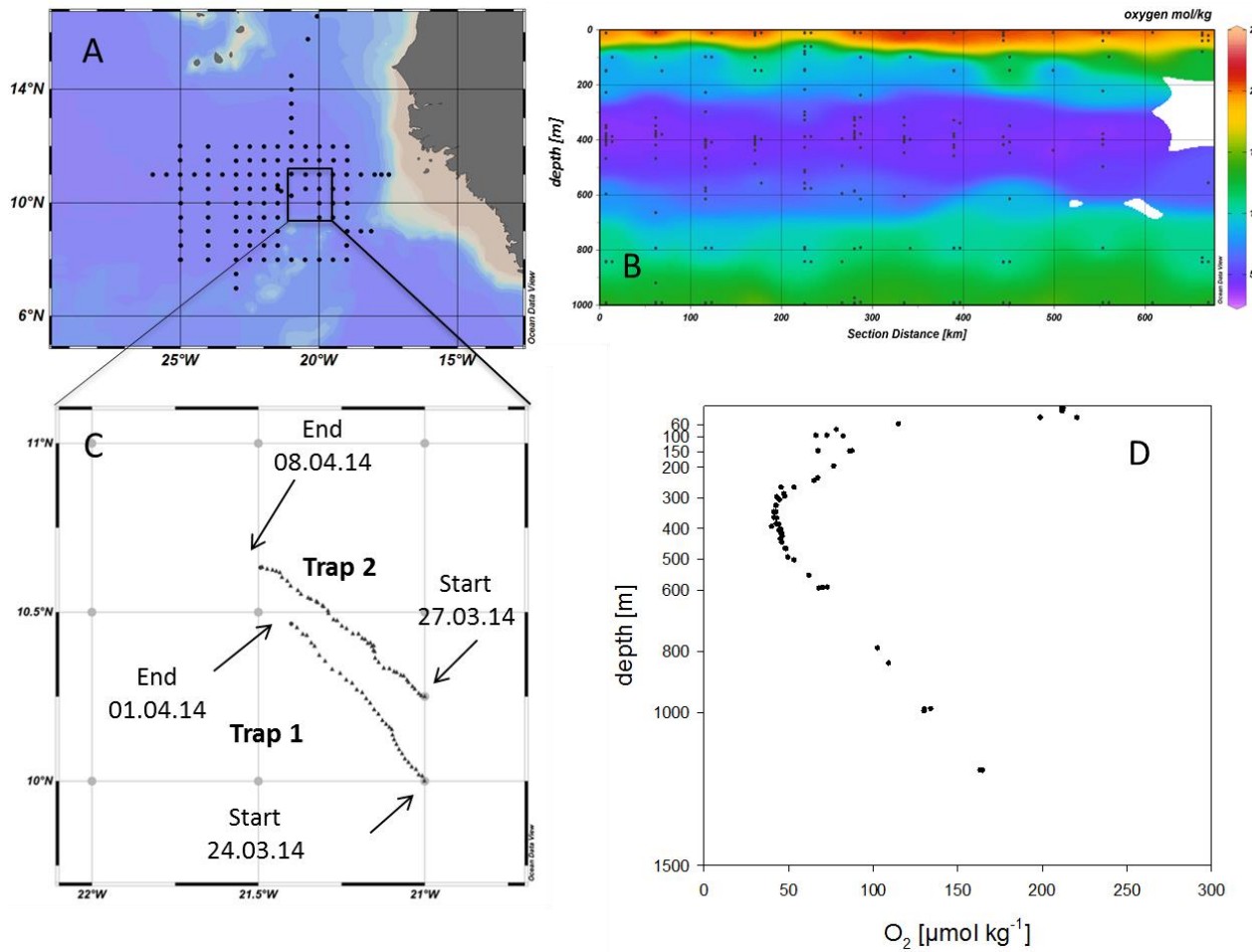





859                 Figure 1a-d


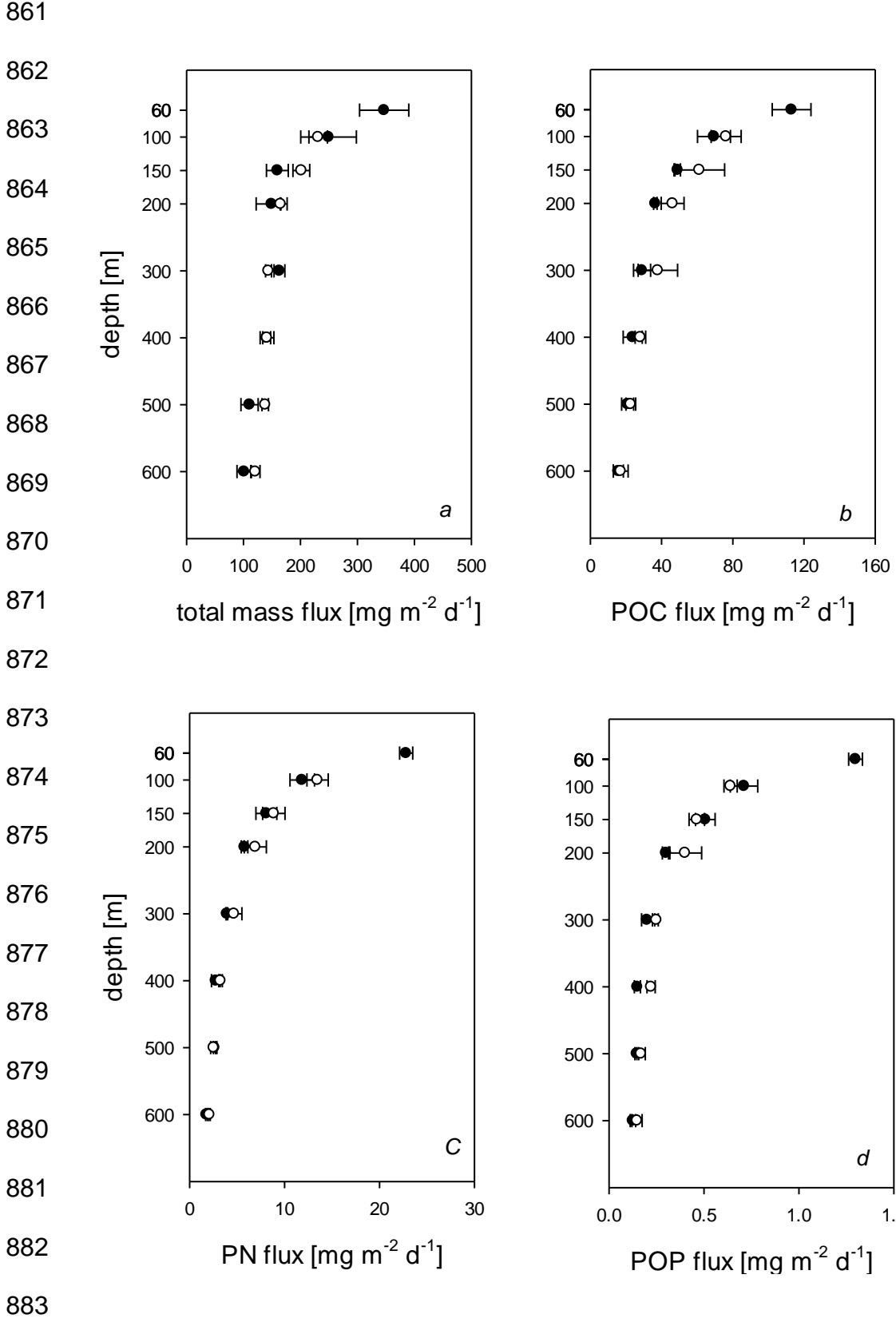

Figure 2a-d

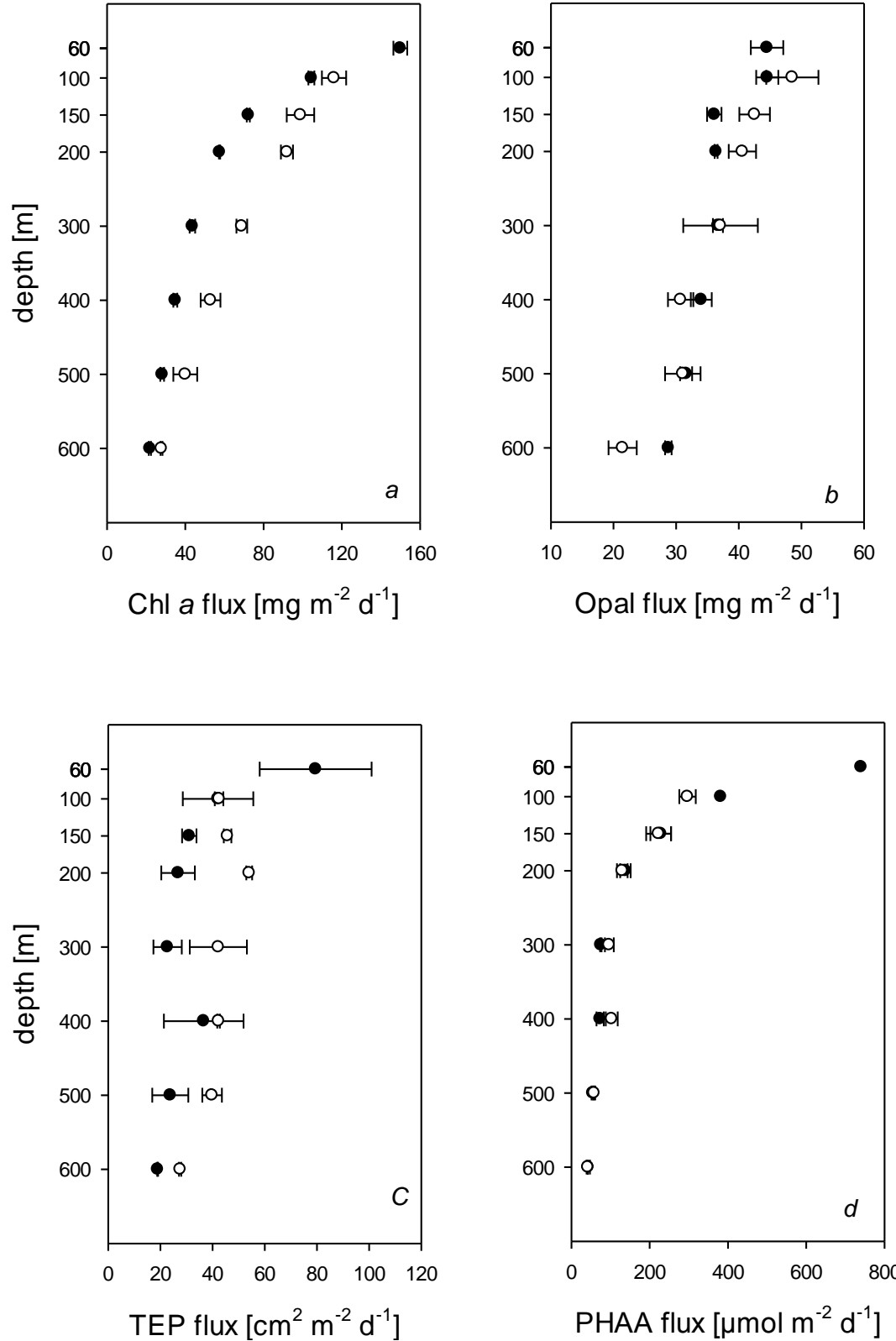



Figure 3a-d

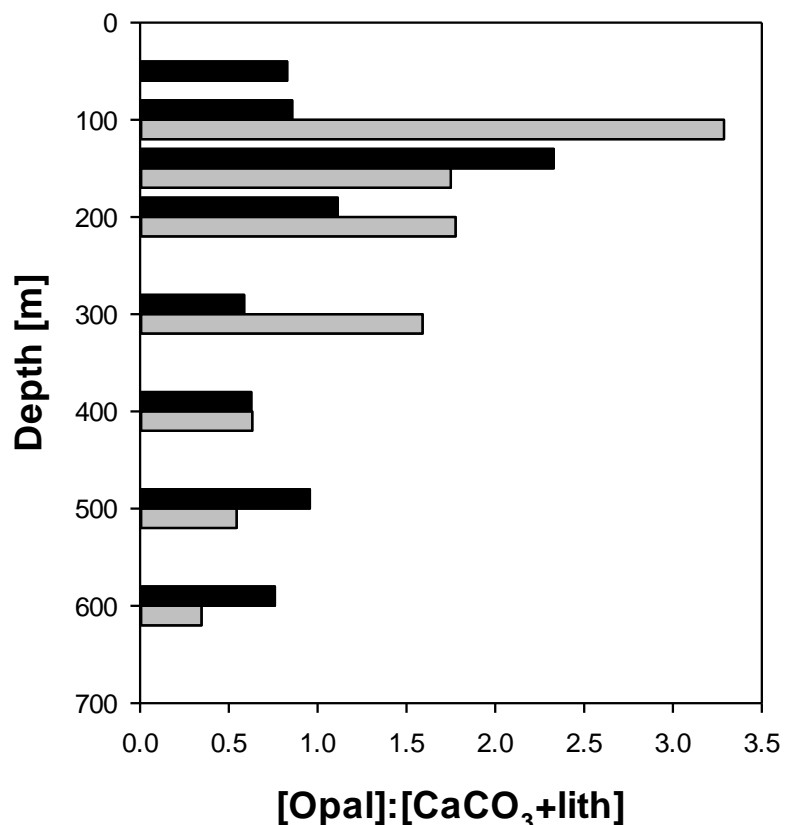

Figure 4

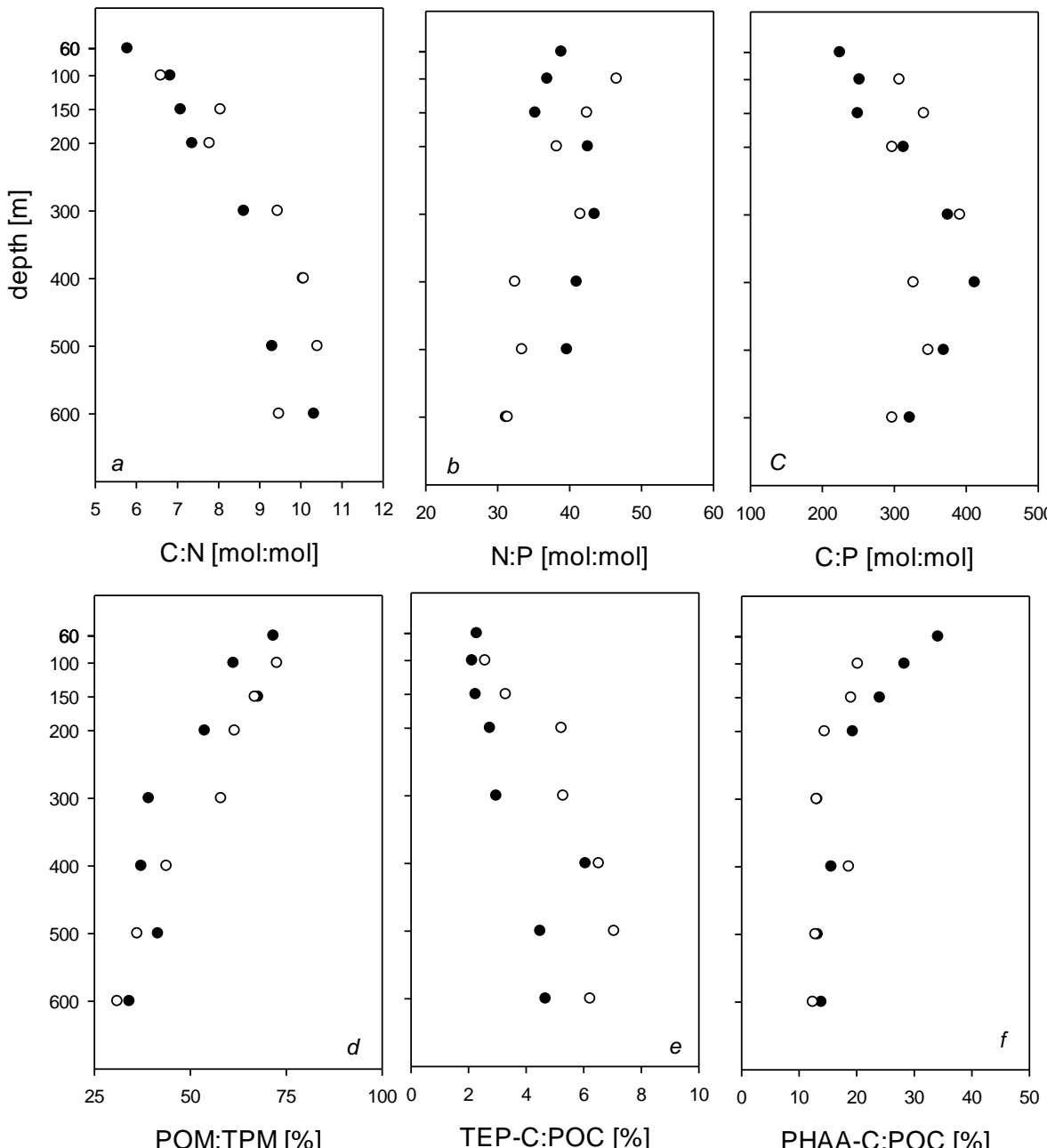



912                              Figure 5a-f






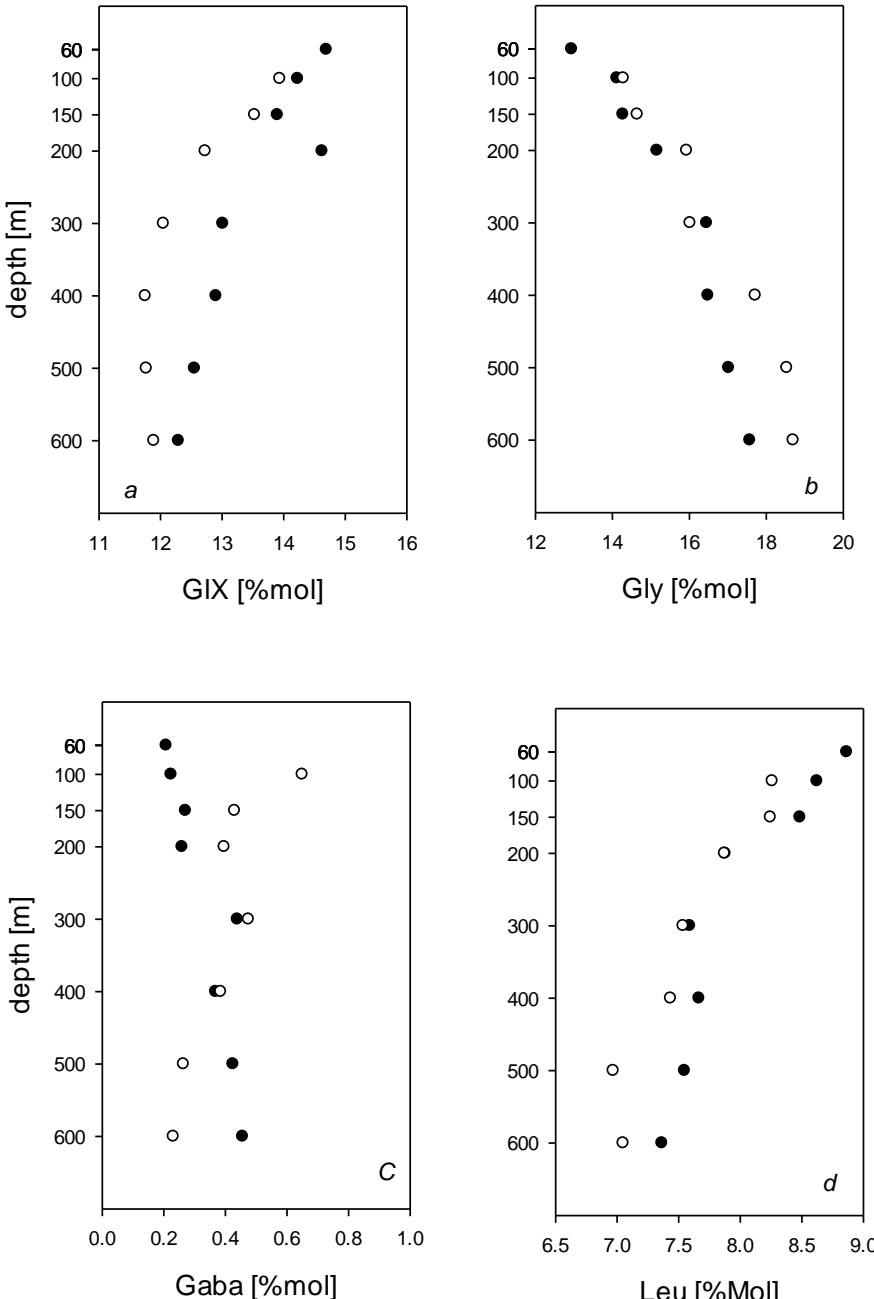




921            Figure 6a-d







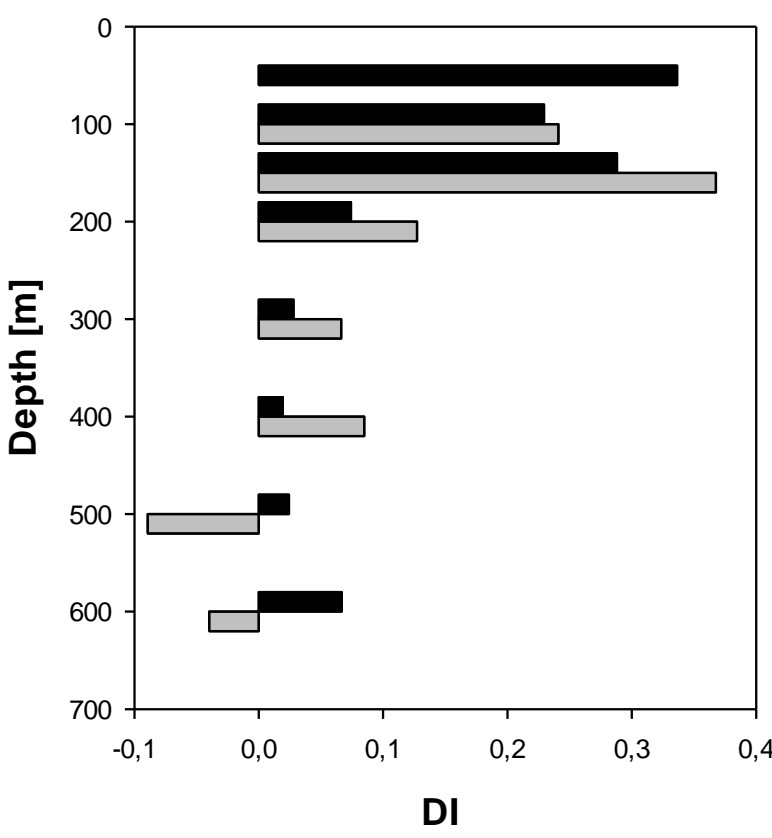



Figure 7