# Peer review of "Particle export fluxes to the oxygen minimum zone of the Eastern Tropical North Atlantic"

_Biogeosciences, 2016_

## Referee Comment (RC1) · Anonymous Referee #1 · 15 Dec 2016

Review of bg-2016-508 Particle export fluxes to the oxygen minimum zone of the Eastern Tropical North Atlantic Anja Engel, Hannes Wagner, Frédéric A. C. Le Moigne, Samual T. Wilson

The authors present a study of vertical fluxes collected with surface tethered drifting sediment trap from the Eastern Tropical North Atlantic. They collected settling material from 7 depths; 60 m, 100 m, 150 m, 200 m, 300 m, 400 m, 500 m, and 600 m. Depth between 300 and 500 m were sampling within the oxygen minimum zone. The main findings in the study was that transfer efficiencies in an oxygen minimum zone were higher than expected when only considering temperature dependency for the microbial degradation of organic matter and that the composition of the organic matter within the settling aggregates had a large impact on the transfer efficiencies. The latter finding was evident through observations of higher attenuation of amino acids compared

to polysaccharide-rich TEP. The manuscript is well written and the date clearly presented. I only have some minor issues regarding the vertical flux of TEP (see specific comments). I recommend the manuscript for publication in Biogeosciences with minor revision.

Specific comments:

Line 57: Please insert a comma after ".. (Volk and Hoffert, 1985)".

Line 210: Were the filters for the elemental analyzer wrapped in tin foil or packed in aluminium cups?

Line 364: The Gum Xanthan flux is per square meter, please correct to m-2 d-1 for both Martin et al. (2011) and Ebersbach et al. (2014).

Line 388-389: It could also be due to slower sinking velocities. It is not possible to say from b alone which process is driving the values. However, you can say that more degradation occurred within a depth region, either due to faster degradation or slower settling.

Line 417-419: Looking at figure 3c, I do not see this trend? For deployment #2 there is an increase above the OMZ, then a slight decrease between 200 and 300 m whereafter it is stable and then show a decrease between 500 and 600 m. For deployment #1 it seems like there is no significant changes in TEP flux between 150 and 600 m. So I do not see that there is a clear different between TEP fluxes within the OMZ compared to below.

Line 464-466: This was only observed for deployment #2, not for deployment #1. Deployment #1 showed decreasing ratios already within the OMZ, 400 to 500 m.

---

## Referee Comment (RC2) · Anonymous Referee #2 · 10 Feb 2017

**Particle export fluxes to the oxygen minimum zone of the Eastern Tropical North Atlantic**

This paper by Engel et al. describes the flux and composition of particles from surface waters through the oxygen minimum zone of the water column off the coast of Mauritania in the Eastern Tropical North Atlantic. Two deployments of surface-tethered drifting sediment traps collected particles at 7-8 depths including the OMZ from 300-500m. Calculated transfer efficiencies of the total POM and various characterized components of the particles indicate lower attenuation through the OMZ than predicted from seawater temperature. Particle composition data showed highest transfer efficiencies for TEP and lowest for amino acids. The paper is well written and a nice contribution regarding particle flux dynamics in OMZs. I have no major issues with the text and recommend publication after minor revisions detailed below.

[Figure]

**Minor comments:**

- pg 3, line 57: add comma after citation
- pg 4, line 81: ref LeMoigne et al., 2012 is missing from reference list; LeMoigne et al., 2016 is listed but not cited, perhaps one of these is an error
- pg 4, line 85: typo, Alldredge is misspelled
- pg 4, line 98: typo, change 'be' to 'been'
- pg 5, line 130 (and elsewhere throughout): I am not familiar with this spelling of 'Mauretania' with an e and could not find other references to Mauritania spelling as such so would recommend changing (certainly if this is an accepted spelling it does not need to be changed)
- pg 5, line 131: 'drifting' is written twice
- pg 6, line 145: hyphenate 'surface-tethered'
- pg 8, line 185: insert 'of' (Aliquots of samples. . .)
- pg 9, line 215: remove 'in' (. . .were filtered onto. . .)
- pg 9, line 221: hyphenate 'peroxydisulphate-containing'
- pg 9, line 231: ref Dittmar et al., 2009 is missing from reference list
- pg 11, line 269: close parentheses after 'sample'
- pg 17, line 432: opal is incorrectly capitalized
- pg 19, line 488: Keil et al., 2016 is missing from reference list (or incorrectly cited here as 2015 is in the list)
- pg 20, line 494: insert 'the' (. . .explain the relative. . .)
- pg 24, lines 592-600: the two Buesseler et al., 2007 references should be distinguished as a and b here and when cited in the manuscript
- pg 26, line 652: Giering reference is formatted incorrectly (begins with 'Sarah')
- pg 27, line 675: Kartensen et al., 2008 is not cited in manuscript but is listed here
- pg 31, line 760: Ploug and Jorgensen, 1999 is not cited in manuscript but is listed here
- pg 34: If possible, it would be nice to include the calculated degradation indices along with the amino acid composition in table 2.

---

## Author Comment (AC1) · 3 Mar 2017

We thank the referee for these suggestions. All of them will be considered in a revised version. Specifically, we will include reduced sinking velocity as an alternative explanation for higher b-values. We will clarify that a decrease in TEP fluxes below the OMZ, and thus a decreasing TEP-C:POC ratio, was mainly observed during deployment #2.

---

## Author Comment (AC2) · 3 Mar 2017

We thank the referee for these suggestions. All of them will be considered in a revised version. Values for the DI's will be included in table 2.

---

## Author Response (AR1)

**Authors' response to referees**

We thank both referees for their supporting comments and good suggestions to improve our manuscript.

During the revision, we recognized a mistake in the conversion of TEP fluxes reported by Martin et al (2011) and Ebersbach et al (2014) into TEP-C fluxes. We corrected this mistake. The change did not affect our interpretations.

Anonymous Referee #1

Review of bg-2016-508 Particle export fluxes to the oxygen minimum zone of the Eastern

Tropical North Atlantic Anja Engel, Hannes Wagner, Frédéric A. C. Le Moigne, Samual T. Wilson

The authors present a study of vertical fluxes collected with surface tethered drifting sediment trap from the Eastern Tropical North Atlantic. They collected settling material from 7 depths; 60 m, 100 m, 150 m, 200 m, 300 m, 400 m, 500 m, and 600 m. Depth between 300 and 500 m were sampling within the oxygen minimum zone. The main findings in the study was that transfer efficiencies in an oxygen minimum zone were higher than expected when only considering temperature dependency for the microbial degradation of organic matter and that the composition of the organic matter within the settling aggregates had a large impact on the transfer efficiencies. The latter finding was evident through observations of higher attenuation of amino acids comparedto polysaccharide-rich TEP. The manuscript is well written and the date clearly presented.

I only have some minor issues regarding the vertical flux of TEP (see specific comments). I recommend the manuscript for publication in Biogeosciences with minor revision.Specific comments:

Line 57: Please insert a comma after ".. (Volk and Hoffert, 1985)".

*Response: comma was inserted*

Line 210: Were the filters for the elemental analyzer wrapped in tin foil or packed in aluminium cups?

*Response: The filters were enclosed in tin cups; information was added.*

Line 364: The Gum Xanthan flux is per square meter, please correct to m-2 d-1 for both Martin et al. (2011) and Ebersbach et al. (2014).

*Response: Units were corrected*

Line 388-389: It could also be due to slower sinking velocities. It is not possible to say from b alone which process is driving the values. However, you can say that more degradation occurred within a depth region, either due to faster degradation or slower
settling.
*Response: Potential effect of reduced sinking velocity was included*
Line 417-419: Looking at figure 3c, I do not see this trend? For deployment #2 there
is
an increase above the OMZ, then a slight decrease between 200 and 300 m
whereafter
it is stable and then show a decrease between 500 and 600 m. For deployment #1 it
seems like there is no significant changes in TEP flux between 150 and 600 m. So I
do not see that there is a clear different between TEP fluxes within the OMZ
compared
to below.
*Response: During both deployments no clear decrease in TEP fluxes was*
*determined for the OMZ. TEP fluxes decreased between 500 and 600m, i.e. below*
*the OMZ; this was more pronounced during the second deployment. We will modify*
*the text to clarify this better.*
Line 464-466: This was only observed for deployment #2, not for deployment #1.
Deployment#1 showed decreasing ratios already within the OMZ, 400 to 500 m.
*Response: Correct, we will modify the text to state that a decrease in TEP-C:POC*
*ratios below the OMZ was only observed during deployment #2.*
**Anonymous Referee #2**
**Particle export fluxes to the oxygen minimum zone of the Eastern Tropical**
**North**
**Atlantic**
This paper by Engel et al. describes the flux and composition of particles from
surface
waters through the oxygen minimum zone of the water column off the coast of
Mauritania
in the Eastern Tropical North Atlantic. Two deployments of surface-tethered drifting
sediment traps collected particles at 7-8 depths including the OMZ from 300-500m.
Calculated transfer efficiencies of the total POM and various characterized
components
of the particles indicate lower attenuation through the OMZ than predicted from
seawater temperature. Particle composition data showed highest transfer efficiencies
for TEP and lowest for amino acids. The paper is well written and a nice contribution
regarding particle flux dynamics in OMZs. I have no major issues with the text and
recommend publication after minor revisions detailed below.
**Minor comments:**
- pg 3, line 57: add comma after citation
- pg 4, line 81: ref LeMoigne et al., 2012 is missing from reference list; LeMoigne et
al., 2016 is listed but not cited, perhaps one of these is an error

- pg 4, line 85: typo, Alldredge is misspelled
- pg 4, line 98: typo, change 'be' to 'been'
- pg 5, line 130 (and elsewhere throughout): I am not familiar with this spelling of
'Mauretania' with an e and could not find other references to Mauritania spelling as
such so would recommend changing (certainly if this is an accepted spelling it does
not need to be changed)
- pg 5, line 131: 'drifting' is written twice
- pg 6, line 145: hyphenate 'surface-tethered'
- pg 8, line 185: insert 'of' (Aliquots of samples: : :)
- pg 9, line 215: remove 'in' (: : :were filtered onto: : :)
- pg 9, line 221: hyphenate 'peroxydisulphate-containing'
- pg 9, line 231: ref Dittmar et al., 2009 is missing from reference list
- pg 11, line 269: close parentheses after 'sample'
- pg 17, line 432: opal is incorrectly capitalized
- pg 19, line 488: Keil et al., 2016 is missing from reference list (or incorrectly cited
here as 2015 is in the list)
- pg 20, line 494: insert 'the' (: : :explain the relative: : :)
- pg 24, lines 592-600: the two Buesseler et al., 2007 references should be
distinguished
as a and b here and when cited in the manuscript
- pg 26, line 652: Giering reference is formatted incorrectly (begins with 'Sarah')
- pg 27, line 675: Kartensen et al., 2008 is not cited in manuscript but is listed here
- pg 31, line 760: Ploug and Jorgensen, 1999 is not cited in manuscript but is listed
here
- pg 34: If possible, it would be nice to include the calculated degradation indices
along
with the amino acid composition in table 2.
*Response: All suggestions have been considered.  Values for the DI's have been*
*included in table 2.*

[revised manuscript text omitted]

#I, grey bars #II.

**Figures**

¶

[Figure]

                              Figure 1a-d

[Figure]

Figure 2a-d

[Figure]

Figure 3a-d

[Figure]

Figure 4

[Figure]

                          Figure 5a-f

[Figure]

                      Figure 6a-d

[Figure]

                          Figure 7